# An Empirical Study for Senior Citizens Using a Customized Medical Informatics System for Dementia Diagnosis and Analysis

**Hsu-Hua Ho [1], Jien-Jou Lin [2], Jia-Qiao Gong [3] and Tzu-Yi Yu [3,\*]**

1   Neurology Department, Catholic St. Joseph's Hospital, Yunlin 63241, Taiwan; wallyho27@gmail.com
2   Department of Business Administration, Soochow University, Taipei City 100, Taiwan; linjj@scu.edu.tw
3   Department of Information Management, National Chi Nan University, Nantou 54561, Taiwan; s106213510@mail1.ncnu.edu.tw
\*   Correspondence: tyyu@ncnu.edu.tw

**Abstract:** The treatment of dementia-related diseases is a global issue. Taiwan is facing a more serious dementia problem due to the combination of an aging society and a declining birthrate. A great portion of healthcare resources has been utilized for dementia among the aged population. In order to understand how dementia develops in rural areas in Taiwan, a cooperated effort between the university and a regional hospital was formed to develop a customized medical information system to collect and track dementia patients. This efficient customized system compiled information on 768 patients with dementia-released diseases. Big data technology and data mining approaches were then applied to analyze the relevant information. Using statistical analysis, we then extracted useful medical findings from the large amounts of collected medical data. Some of the findings indicate that the patients' education level and care practices have a major effect on the dementia severity in these local senior populations.

**Keywords:** healthcare sustainability; medical informatic system; big data





## 1. Introduction

Human beings now live longer and healthier lives than in the past, benefiting from advances in technology, medical equipment, and sanitation. According to the National Center for Health Statistics, which is part of the Centers for Disease Control and Prevention, the average life expectancy in the United States in 1900 was 46.3 years for males and 48.3 years for females. In 2018, the average life expectancy for men and women of all races in the United States was 76.2 and 81.2 years, respectively. The US life expectancy was 78.7 years, up 0.1 years from 2017 [1]. According to a US CDC data study, there was a 20-year rise in average life span expectancy in the second half of the twentieth century; the world population's median age is rising. These factors, together with high fertility rates in many nations during the two decades after World War II, have resulted in a rise in the number of persons aged 65 and older between 2010 and 2030. The average life expectancy is predicted to rise by another ten years by 2050. The growing number of elderly people will place a burden on public health systems, as well as medical and social services [2]. As people are living longer lives, the incident of aging becomes a worldwide issue.

Aging populations are a problem in Asian countries as well as Western ones. Japan's life expectancy in 2019 was 84 years, whereas Taiwan's was 80.86 years [3]. It should be beneficial for individuals to live longer lives than they did in the past. However, diseases which are prevalent in the elderly continue to emerge as individuals live longer lives. According to the World Health Organization, an "aging society" has 7% or more people aged 65 years or older, an "aged society" has 14% or more, and a "super-aged society" has 20% or more, respectively. Taiwan and Japan are two Asian countries with



major age-related problems. During the mid-twentieth century, the majority of Taiwanese inhabitants were youthful farmers. Approximately 2.5 percent of the population was above the age of 65. However, tremendous industrial expansion resulted in a considerable shift in the age demographics of the population. According to the Ministry of the Interior's population strategy white paper issued in 2013, Taiwan became an aging society in 1993, when more than 7% of the population was 65 or older. In 2018, Taiwan became an aged society, with over 14% of the population aged 65 or older [4,5]. Furthermore, the National Development Council issued an alarm in 2014, stating that Taiwan's community had progressed from "aging" to "aged", and that by 2025 it would be a "super-aged" society, with 20~25 percent of individuals being older citizens [4]. The possible causes of speedy aging in Taiwan include medical breakthroughs, a cleaner environment, and advances in science and technology, which increased average life expectancies. A low birthrate might be another hypothesis. This aging issue is not exclusive to Taiwan; it has also occurred in other Asian nations such as China and Japan. The research indicates that China's senior population will continue to expand for decades. The aging problem in China is so severe that it will require a coordinated effort by gerontologists, healthcare professionals, lawmakers, and societal forces [6]. The National Institute of Population and Social Security Research (IPSS) of Japan predicts the present 120 million population will drop to less than 60 million in 100 years. Such severe population shifts affect government economic strategies and different industry areas [7].

Living longer should be a blessing for the citizens. However, aging is frequently accompanied by several comorbid chronic conditions. Some acute diseases, such as heart attacks or pneumonia caused by COVID-19, are fatal and can kill senior citizens quickly [8], and other senior citizens may suffer from those diseases chronically. The disease's lifespan may be quite long, and people who suffer from it may require medication for the rest of their lives. The long-term care and treatment of the elderly have a long-term impact not only on their quality of life but also on their family and care institutions. The frightening aspect of chronic diseases is not just one person's suffering but also the long-term care process, which can deplete the entire family physically and mentally. Alzheimer's and Parkinson's disease are two of the most devastating and agonizing elderly illnesses. These conditions are also referred to as dementia. Dementia has many different symptoms. Dementia is a steady decline of cognitive function caused by brain damage or disease, and the disease has a far harsher impact on patients' brain states than normal aging does [9]. According to the World Health Organization (WHO) guidelines on dementia [10], "Dementia is a major cause of disability and dependency among older adults worldwide, affecting memory, cognitive abilities, and behavior, ultimately interfering with one's ability to perform daily activities. The impact of dementia is not only significant in financial terms, but also represents substantial human costs to countries, societies, families, and individuals [11]."

According to the 2019 Global Dementia report of ADI (Alzheimer's Disease International), an estimated 50 million people are living with dementia. By 2050, the world population will have risen to 152 million people. According to the WHO [10], a new dementia patient is diagnosed every three seconds. The WHO also stated that the global population of dementia patients was nearly 50 million in 2017, and by 2050 the number will reach 131.5 million. It is estimated that the global cost of care for dementia in 2015 was USD$818 billion, and it will exceed USD$1 trillion by 2018 [12,13]. The World Health Organization, foreseeing the serve problem that dementia will pose, has presented the "Global action plan on the public health response to dementia 2017–2025", which aims to improve the lives and jobs of people with dementia, and aid their families. It is also working to rouse public attention on the issue in order to decrease the burden of dementia on communities and countries all over the world [14]. Because it is a global issue, the status of dementia in Taiwan is also very significant. According to information released by the National Health Insurance Administration of Taiwan, part of Taiwan's Ministry of Health and Welfare [15], based on the demographic data of July 2017, there are 3,192,477 citizens over 65 years old in Taiwan. Of this group, 586,068 people have mild cognitive

impairment (MCI), accounting for 18.36% of citizens over 65 years old, and 253,511 people in the population have dementia, accounting for 7.94% of citizens over 65 years old. There is one dementia patient for every 13 people over 65 years old, and one dementia patient for every five people over 80 years old. The statistics report also indicated that

> . . . the population with dementia will exceed 270,000 by the end of 2017; in 2031, the population with dementia will exceed 460,000, and by then every 2 out of 100 Taiwanese will have dementia; in 2041, the population with dementia exceeded 660,000, and there will be more than 3 dementia patients per 100 Taiwanese; in 2051, the dementia population will exceed 810,000, and there will be 4 dementia patients per 100 Taiwanese; in 2061 The dementia population will exceed 850,000, and there will be nearly 5 dementia patients for every 100 Taiwanese [16].

In the next 45 years, the number of people with dementia in Taiwan will increase by an average of 36 people per day—a rate of one new case of dementia every 40 min. This information implies that dementia will put heavy financial pressure on individual families as well as the entire society. As matter of fact, dementia has significant economic and social consequences in terms of direct medical and social care costs, as well as unofficial treatment costs [10,12]. According to the WHO, the total global societal cost of dementia was estimated to be USD$1.3 trillion in 2019, and this cost will be expected to exceed USD$2.8 trillion by 2030 as both the number of people living with dementia and care costs rise. Caretakers spent an average of 5 h per day providing care for individuals with dementia in 2019 [12]. Physical, emotional, and financial pressures can put a strain on families and caregivers, necessitating assistance from the health, social, financial, and governmental fields. Informal care accounts for 50% of the global cost of dementia [10]. Because dementia is not as deadly as most cancers, the suffering and pressure last much longer, which can be very overwhelming for dementia patients and their caretakers. How to detect and track dementia patients efficiently so that the appropriate treatments can be provided is a critical issue in Taiwan.

This research is a collaboration between academic institutions and hospitals. The history and purpose of St. Joseph's (Catholic) Hospital can be utilized to describe the objective of this study. The hospital was founded in 1955 in Huwei Township by Bishop Thomas Niu to provide comprehensive medical care to the citizens of Huwei and nearby areas, particularly the impoverished. He noticed that the people in the town lacked medical care facilities and had to pay unaffordable medical bills, so he invited two foreign missionaries, Fr. George Massin and Fr. Anthony Pierrot, to establish St. Joseph's Hospital based on three principles: "total sacrifice, unconditional love, and constant joy". This hospital has served the economically disadvantaged residents of this small town of approximately sixty thousand residents for sixty-seven years. Many elderly people seek diagnosis and treatment at the neurology department of this hospital. Thus, the residents of this town are the subject of this study. In this work, we first established a tailored medical information system to detect and track dementia patients and their potential risk factors, and then we utilized data mining technologies on these acquired data to gain useful information.

Data mining is the technique of analyzing large datasets to uncover patterns and correlations that can be used to address problems via data analysis. It can help businesses forecast future trends and make informed strategic decisions, even in spite of the fact that there are still certain downsides and concerns surrounding this technique. Data analytics is a complex process that frequently necessitates the use of data mining tools by people who have received training. Furthermore, there are several methods for the analysis of data, some of which are more accurate than others. Security and privacy problems are also mentioned as important drawbacks. One disadvantage is that successful data mining necessitates large datasets [16–18]. However, data mining's usefulness is showcased in an apocryphal anecdote. This intriguing story says that a study of consumers' buying behavior at a supermarket discovered that men frequently purchased beer and diapers together during weekend football games. Following this observation, the management intentionally positioned beer and diapers closer together, resulting in a significant rise in sales. Although

the validity of this legend cannot be proven, discovering correlations between potential risk factors for dementia diseases is our essential research approach. The Pearson Chi-square test was the major tool we used to analyze the collected data. This method has been used widely to determine the possible relations between two variables [19]. Although the Chi-square test determines likely relationships between two variables, it does not offer enough information regarding the relationships between one and others. It has also been criticized for not being able to quantify the relationships. Fisher's exact test is an alternate solution that is used not only to determine if there are nonrandom correlations between two categorical variables but also to examine the significance of the link between the two types of categorization. However, the Chi-squared test is an approximation for a large sample, whereas Fisher's exact test is a precise approach, particularly for small samples. This statistical method uses the calculation of factorials. In addition to requiring sufficient processing power to compute the factorials of big numbers, the approach will induce numerical overflow. As previously stated, data mining may obtain correct information from a large number of samples; nevertheless, a large number of samples surpasses the calculation range of Fisher's exact test [20]. Because of this quandary, this study continues to analyze data using the widely utilized Chi-square test. Like the majority of other research and reports [3,10,11], ours does not compare the significance of the risk factors. The goal of this study is to identify dementia risk factors for this town's residents and take the appropriate action to enhance their health.

This paper is structured as follows. Section 2 discusses dementia-related literature, big data approaches, dementia diagnosis and the need for tailored medical software. Section 3 describes the software's techniques, implementations, and dementia patient information. Section 4 presents our hypothesis and data analysis. Section 5 discusses the study's contribution and limitations. Section 6 concludes with summarizes and future works.

## 2. Medical Background and Approaches

This study does not intend to explore new drugs or medical therapies for dementia. Instead, we concentrate on identifying dementia risk factors using appropriate procedures, and on the response strategy used to improve the patients. Research related to dementia has been widely discussed. Dementia is mainly caused by damage to or the loss of nerve cells and their connections in the brain [9,10]. Because there is no complete cure for most of the diseases related to dementia, early detection is important. Dementia has many risk factors (such as high cholesterol, smoking, or alcoholism) which might trigger the causes of dementia. If these relevant risk factors can be detected early, dementia can be prevented or slowed down. If we could use the proper methods to know the relationship between different risk factors and the severity or types of dementia, it would be very helpful to public healthcare. This study uses big data analysis to determine the risk factors of dementia and the relationship between risk factors and disease severity. Big data approaches to healthcare are very common practices [17,18,21]. Due to the development of science and technology, the amount of data to be processed has increased. When there is a large amount of data, one can't just look at it to know its value, or to know how certain elements are related to each other. Whether the value of the "big data" and other data affect each other depends on the statistical methods used to find the "hidden" information in the data. This hidden information can be used to formulate policies or solve problems, and this is the most important and valuable part of big data analysis. The procedures of big data analysis can be summarized as follows: the first step is data acquisition, which refers to the collection of a sufficient amount of data from the same source. The second step is to format the collected big data; this involves deleting inconsistent, duplicate, useless, missing, or conflicting data. The third step is the data analysis, which selects appropriate methods for the statistical analysis of the data according to the characteristics and demand purpose of the data. The fourth part is data mining, which further excavates the analyzed data to find out the effective information hidden in the data. The last step is information interpretation and analysis, which involves interpreting the results obtained after data

exploration, and then presenting the resulting reasonable assumptions and conclusions so that those acquired data can become useful information [22]. This research tries to follow these steps and make useful suggestions for decision-makers to define policies which are suitable for the elderly citizens in a rural town.

The process of diagnosing dementia can be described as follows: When patients arrive at the hospital, they undergo general examination procedures to determine whether the patients are suffering from either general aging or dementia. The doctor's inquiry and initial prescription are the first steps. The physician's investigation comprises a preliminary examination, an inquiry into the patient's current symptoms and prior drug usage, and a review of the patient's past medical history. Next, the patient undergoes a neurological examination based on their symptoms, including a test to determine whether there is combined limb weakness, recognized Parkinson's disease symptoms, and so on. Blood tests—such as standard blood tests, liver and kidney function assessments, electrolyte problems, vitamin B12 concentration, folic acid level, thyroid function, syphilis serum testing, and so on—are also included in the assessment procedures. Brain imaging examination techniques, such as CT (Computed Tomography) or MRI (Magnetic Resonance Imaging), will be used to determine the patient's conditions if more testing is required. These methods can evaluate brain atrophy and rule out other intracranial abnormalities. An electroencephalography (EEG) test can also be utilized to assess the patient. Additionally, a cerebrospinal fluid biomarker analysis or another nuclear medical investigation may be used. On the other hand, dementia is frequently diagnosed using methods for the assessment of cognitive function. These instruments evaluate patients' cognitive abilities and establish the severity of their dementia using a variety of scores. The two testing scales used most often are CDR (Clinical Dementia Rating Scale) and MMSE (Mini Mental State Examination). Usually, these general examination processes may be completed in a month, but this also depends on how frequently patients return for follow-up appointments. The type of dementia the patient has and its severity may be clearly confirmed after six months of assessment and inspection.

These examination procedures will generate a wide range of data, including digital data, picture data, a brain wave map, or a diagnosis text. These data will be dispersed throughout the hospital's various databases, hospital-purchased examination equipment, and software from commercial firms. Among the purchased software, the medical information system is commonly utilized by doctors to document patients' consultations. Most physicians in different departments (such as gynecologists, pediatrics, otolaryngologists, etc.) use the general medical information system to diagnose patients' problems and prescribe medication. They write the patients' chief complaints and symptoms in a text format and save these statements in the hospital's database. The diagnosis report will also be written by the physician at the end. Finding the patients' wide range of inspection information quickly from various databases using the general medical information system is quite challenging. Most commonly, doctors and specialists type diagnostic reports in various languages (such as English or Chinese). English acronyms will also be used in place of some terms. Even the same disease may have several distinct acronyms used by various doctors. Additionally, grammatical errors and typos can commonly occur (like the misspelling of Alzheimer's). All of these make it challenging to monitor and identify dementia patients using the query function from the general medical information system. Finding the required dementia patients and other medical histories from several databases is a time-consuming and laborious task for the big data analysis of dementia. Creating a more personalized medical information system can resolve this issue.

## 3. Methods and Implementations

This tailored medical information system, which serves as a front-end platform to collect patients' information, does not use mathematical formulae. Instead, the crucial components are an experienced physician with neurological expertise and a skilled software development team. Physicians should understand not only what examinations are

necessary, but also what processes should be built into the medical information system so that physicians may utilize it straightforwardly and efficiently throughout the patients' consultation. A competent software development team will create precise database architecture for future dementia patient tracking and inquiries. Previously, physicians typed the words for patients' diagnosis reports; this program adds user interfaces to allow them to check proper items in addition to the verbose writing. This capability allows dementia information to be precisely recorded in the database in order to prevent those drawbacks previously mentioned. The system may then export these patients from our databases in the required search criteria when we have collected a sufficient number of patients, and the data analysis and mining methods can be applied to these data to obtain the hidden correlation information. Because most hospital medical equipment is linked to Windows PCs, this neurology-specific diagnostic system uses Windows OS. After selecting the framework, the window-based database was chosen. Software engineering was applied throughout the design and development cycle. During project implementation, attending physicians' or nurses' advice was used to construct the GUI. The relational database was chosen so that it could be administered by the database management system (DBMS) and synchronized with the original hospital database and other existing systems, reducing development costs and time. In order to fit as many functions in the display window as possible, we utilized the "Tab" entity. Each Tab's display and diagnostic features are different. Figure 1 depicts this customized medical system. Patient information is withheld for privacy (the same procedure is applied to the other figures).

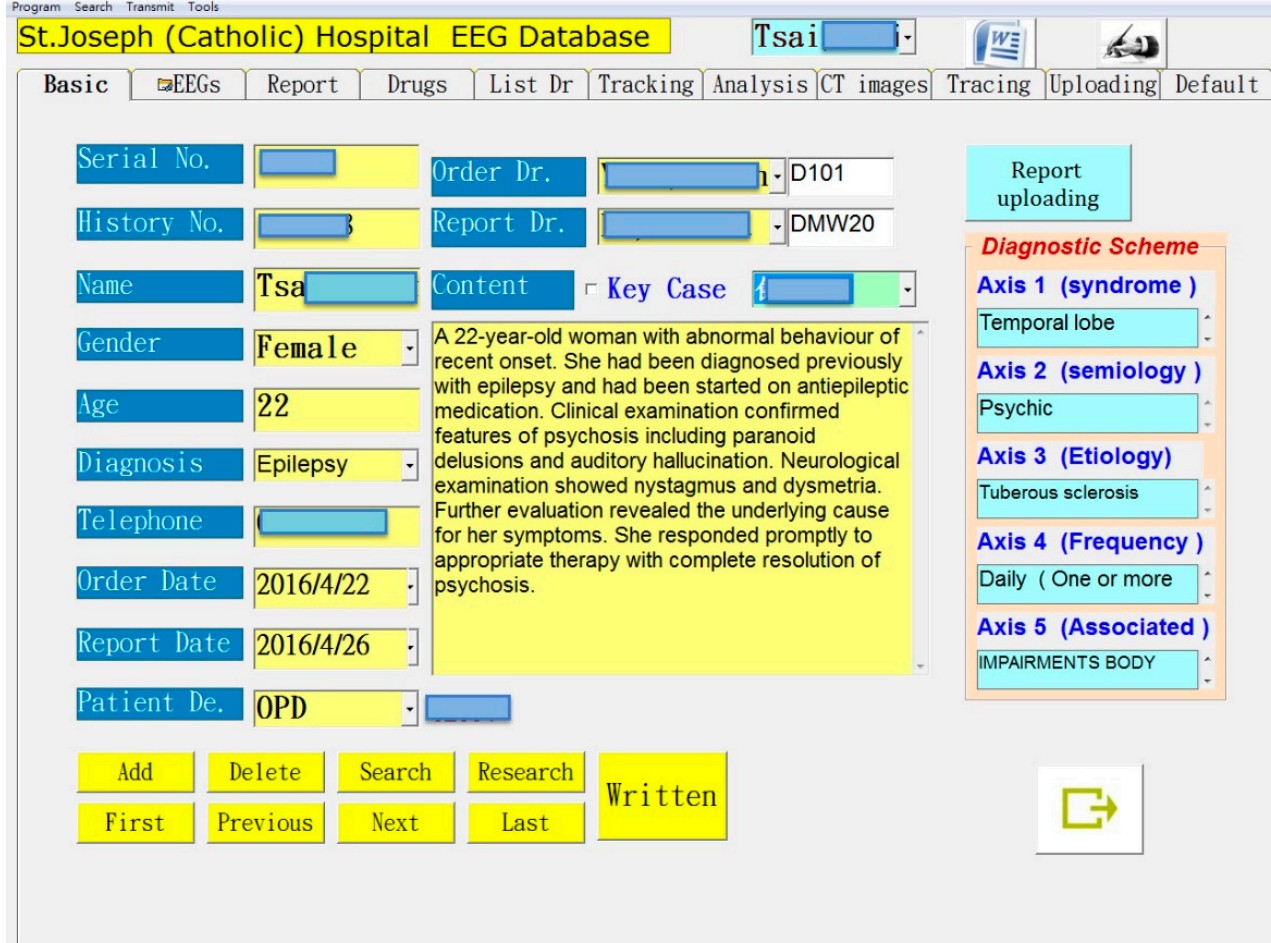

**Figure 1.** The customized medical information software system.

*3.1. Customized Designed Medical Information System*

Typing is the most popular user interface for the general medical information system. As we have already mentioned, typing in the diagnosis system could cause inefficient and inaccurate issues. We utilize checkboxes and listboxes to let doctors "check" if a patient has a dementia-related disease. The physician may simply determine the type of dementia the patient has by selecting the appropriate CheckBox or ListBox item. This design overcomes previous problems in the general medical system. Searching a database with text querying is challenging and inefficient, as the initial diagnosis statement may not be precise. The checked function provides an easy way to identify dementia patients. Our system incorporated statistical approaches for the assessment of patient data, for example, in order to identify the particular factors for other diseases. This system uses a plug-in library to produce data in the Office format (such as Apache OpenOffice or LibreOffice). Office file formats are popular and easy to use. This program not only enables doctors to write diagnosis statements but also includes fundamental query functions. Patients with certain diseases can be located using the diagnostic disease query option. After the inquiry, patients with a specified ailment are presented. A physician can also establish age, registration number, and illness thresholds. Using logical operators (AND, OR, Less Than, Greater Than, etc.) and a range helps to improve the query. This diagnosis system's logical operators are unique compared to the fundamental system utilized by physicians or internists. For example, if physicians want to know the details of female patients who are uneducated, under the age of 70, and have Degenerative Dementia or Vascular Dementia, they may use these specially designed logic operators to set the requirements. The software will create a SQL query string such as "SELECT * from dementia.table WHERE Gender = 'FEMALE' AND Age $\leq$ 70 AND (Dementia = Degenerative OR Dementia = Vascular)", then pass this string to query the table of the database (presume the name of the data table is called dementia.table). If the query command is successful, the query results will be displayed for the physician to investigate. The relational database query can also be used to obtain information from tables in different databases. If advanced statistical approaches are needed, our program may export data in LibreOffice Calc or Excel for additional analysis (for instance, the SPSS or SAS). These query functions were previously carried out by activating other database management software and performing extra operations. Nonetheless, these activities are now incorporated into this customized medical software.

This system can create LibreOffice Write or Word file format documents for patient reports that may be printed or saved electronically. This system was developed for neurology departments, and can incorporate common symptoms and doctor's prescriptions, especially for neurology. The most common complaint from patients about their diagnosis reports, that they were written in English with medical terminology and were difficult to understand, would then be resolved. This unique feature is vital for patient referrals or follow-up appointments with other doctors. This physician-oriented method makes it easier to obtain more accurate and efficient information about dementia patients than the other general system utilized by most physicians in other departments. Figure 2 shows the snapshot of this software.

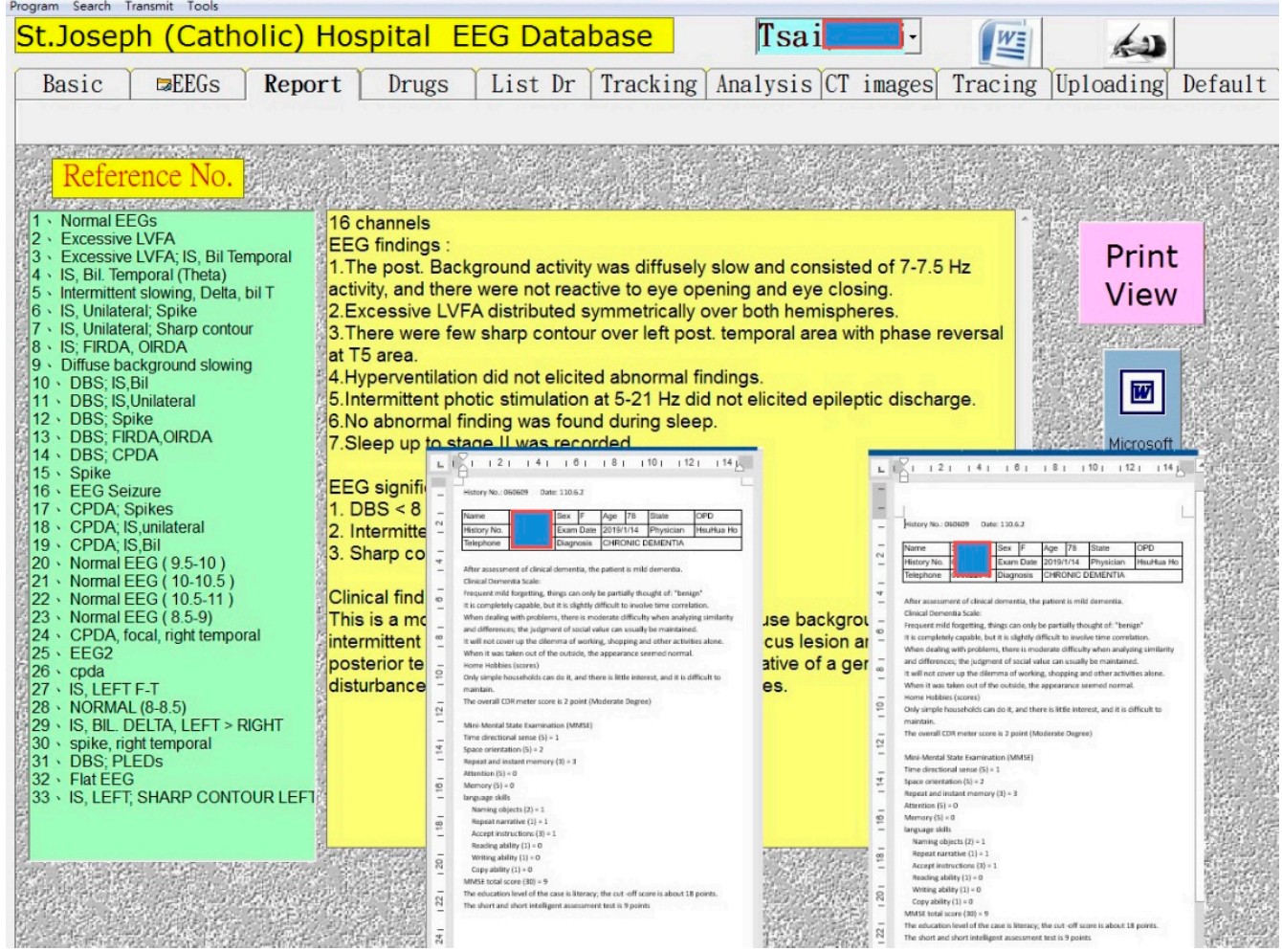

**Figure 2.** Seamless integration with OpenOffice file output.

### 3.2. Versatile Functions Provide the Decision Support Capability

Many examining methods can be used to determine whether a patient has a dementia-related disease. A common method is clinical observation and querying. There are also tools to evaluate the mind's health condition, such as the AD8 (Eight-item Informant Interview to Differentiate Aging and Dementia), SPMSQ (Short Portable Mental State Questionnaire), MMSE, CDR and CASI (Cognitive Abilities Screening Instrument) forms. The AD8 is an eight-item questionnaire which assesses memory, orientation, executive functioning, and interest in activities. The SPMSQ is a ten-item questionnaire, and is administered by a clinician in an office or in a hospital to measure a patient's mental status. The MMSE is the most commonly used test for people who might be worried about memory problems or dementia and its progression. CDR is a global rating scale for the assessment of patients diagnosed with dementia. The CDR form evaluates the cognitive, behavioral, and functional aspects of Alzheimer's disease and other dementias. The final tool is the CASI, which provides a quantitative assessment of attention, concentration, orientation, short-term memory, long-term memory, language abilities, and visual construction for mental health evaluation. The detail of these assessing tools can be found in the documentation [10,11]. The questionnaires are provided on paper when they are used to evaluate patients. Traditionally, nurses or medical faculties will use these assessing forms to evaluate the patients' dementia severity. The results (the scales and scores) are then recorded in their database. This procedure is generally tedious and laborious. Because the contents of those questionnaires can be retrieved from the internet, it is easy for dishonest people to cheat on tests for certain illegal purposes and lie about their personal information. We implemented

electronic versions of these assessment tools. The questionnaires are displayed randomly on our software display window to prevent people from remembering the associated answers. Some questions have to show in proper sequence (for instance, question b has to pop up after question a); these questions will be grouped to ensure the order sequence has been maintained. This function facilitates the evaluation process, and the associated test scores are recorded in the database automatically after the assessment measure is completed. The random popup questions feature makes the evaluation more objective, and is much more effective than the one using hard copies. This is demonstrated in Figure 3.

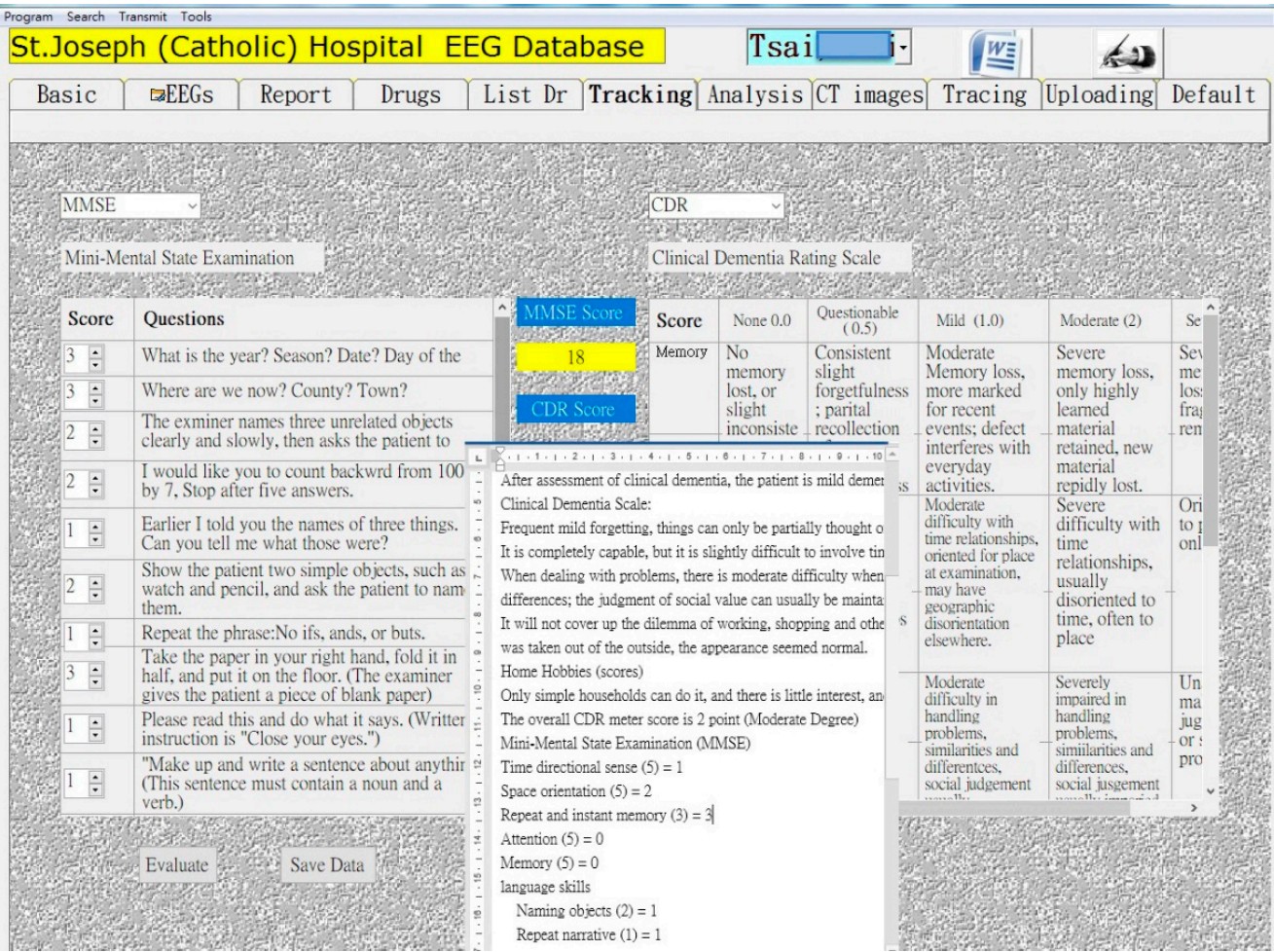

**Figure 3.** Dementia assessment tool implemented in electronic format.

Furthermore, if the resulting scales of different tests (such as the CASE, CDR, and MMSE) are not on a similar level, the assessment results cannot accurately reflect patients' real conditions. In order to amend this kind of inconsistency, our system provides another reference to verify the patients' real condition. As we previously mentioned, most dementia-related diseases occur because of brain injury or brain-related disease, and these physical lesions can be precisely detected by MRI/CT or EEG (electroencephalography, commonly refer to the brain waves). MRI and CT are very common approaches used for brain imaging. It is very important to provide enough information for physicians to diagnose the patients' actual mental condition or the severity of their condition. For other physical exam methodologies, such as MRI, CT, or EEG, the results can also be loaded into the system and displayed concurrently in a different diagnosis window, in addition to those scales of the assessment tools. The examining images can provide strong evidence to support a physician's decision. This self-developed software helps the physician to perceive patients' history records or examination results without switching between systems. More

patients' data along with strong decision-support information can be provided to the physicians in order to assist them in making more proper diagnoses. If doctors need more specific guidelines or details to back up the diagnosis, they can open a context menu by right-clicking the mouse and selecting the link to Radiologic Anatomy for more information. This decision support function improves senior healthcare assessment and treatment significantly. These functions of our self-developed software are shown in Figure 4.

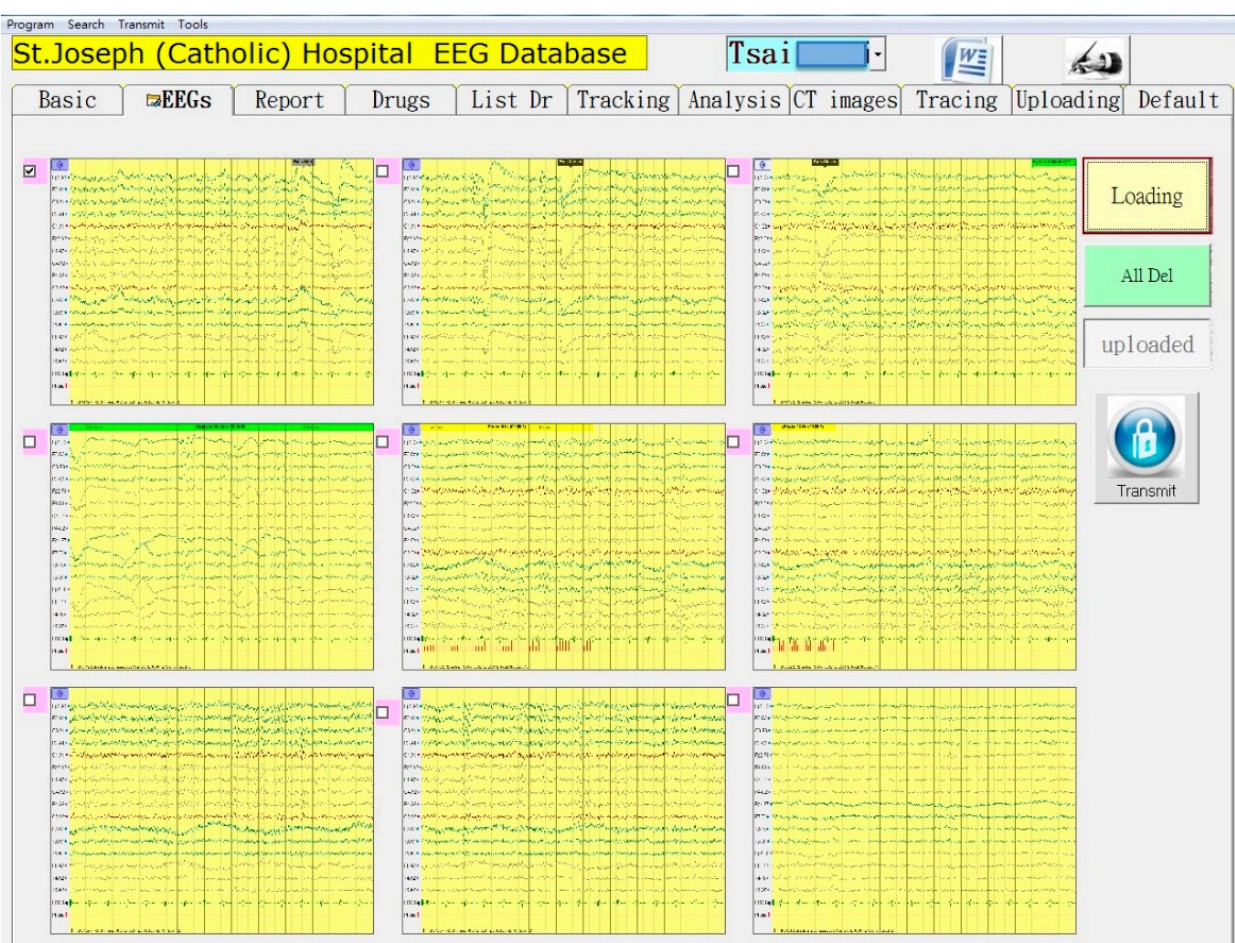

**Figure 4.** The self-developed system allows access to the EEG database to assist in diagnosis.

### 3.3. Software Engineering Development Cycle

In hospitals, many different systems and types of software are used. Efficiency and effectiveness issues will arise if they are not properly integrated or do not follow the hospital's standard protocol [23]. Nurses, as well as other hospital personnel, may have to put in more effort to adjust to the unintegrated systems. This generally leads to increased nurse and faculty burnout while also jeopardizing patient safety. Our customized neurology diagnosis medical system adheres to the software development cycle in order to ensure that it can complete work with all functions implemented at maximum capacity in the hospital, thus benefitting the patients. The design life cycle of an information system is a project management concept that describes the stages involved in developing an information system, from the initial feasibility study to the ongoing maintenance of the finished product. It follows the pattern of most new discoveries: identify problems, plan to assess a solution, evaluate and design a solution, and then establish and monitor the solution. Many good project advancements have been noted using the system design life cycle procedure [24]. At the end of May 2017, Chen Shizhong—the commander of the National Health Command Center of Taiwan—led a delegation to Geneva, Switzerland to participate in the drafting of

the "2017–2025 Global action plan on the public health response to dementia" approved by the World Health Organization [14]. He attached great importance to this dementia problem among aged people, and proposed seven national-level policies, one of which was to build up the medical information system for dementia. Since then, the Catholic St. Joseph's Hospital's neurology department has cooperated closely with us from Chi-Nan University. The development team decided to use a personal computer running the Microsoft operation system as the workbench. After the workbench was determined, the database used under the windows operation system was chosen. This self-developed program advanced through several levels. It utilized the Microsoft Access databases and Visual Basic initially. Later, the front-end application was embedded with VBA (Visual Basic Applications) and the back-end database was an SQL server. After Microsoft released a new operating system and software, this system embraced the benefits of the continuous update function, resulting in improved reliability under the Microsoft.net framework. Even during the development and design cycles, conventional software engineering methodologies were used. The database design must be prepared for execution in a relational database so that it can be managed by a database management system. This procedure was implemented to integrate this homemade software with the hospital's original Hospital Information System, and to reduce development expense and duration. Many explanation and training sessions were held during the deployment and disposal stages in order to advertise this self-developed dementia record-keeping system. The development team reviewed all of the feedback and amended and debugged the system to ensure that it met the requirements of a reliable piece of software. A cloud hard disk with an Na AS system is used for automatic periodic backup to secure the database system. This hospital's self-developed system has been operational for over ten years.

*3.4. Data Acquisition and Formation*

The first step of big data analysis is to obtain the correct data for investigation, and then the hidden information in the data can be used to provide a more complete understanding of the disease. Research in Germany has shown that this procedure is important [25]. The authors collected long-term patient data from clinical facilities because they believe that data is one of the most important prerequisites to providing them with a better understanding of Alzheimer's disease. This is a very common approach, and is also the reason behind applying big data analysis to healthcare problems. Because the physician-oriented diagnosis software for dementia was implemented in St. Joseph Hospital, it has been used to collect patients' data in the hopes of understanding more about how risk factors affect citizens with dementia. St. Joseph's Hospital is a catholic hospital located in a town with approximately sixty thousand people, where many young people moved out to metropolitan centers for a living. In this town, the ratio of elder citizens living alone or staying with foreign nursing workers is relatively high. Even though the possible risk factors which cause dementia have been presented [9,12,26], in order to better understand from our clinical experience of what risk factors are most related to dementia in our residents, we used this customized medical information software for several years to collect dementia-related data. There were thirty items tracked for each patient. They were the patients' age, gender, education level, current care methods used (staying with family or using a foreign care worker), living area (which might be related to water quality), hypertension, hyperlipidemia, diabetes, and cerebral apoplexy (stroke). The others are the physical examination data, such as the blood urea nitrogen (BUN), creatinine (CREAT), glutamate-oxaloacetate transaminase (GOT), glutamate-pyruvate transaminase (GPT), sodium (NA), potassium (K), a venereal disease research laboratory test (VDRL), vitamin B12, thyroid-stimulating hormone (TSH), free thyroxine T4 (FT4), hemoglobin (HGB), total cholesterol (TCHOL), triglycerides (TG), high-density lipoprotein (HDL), low-density lipoprotein (LDL) and blood glucose (Glu). These physical examinations may be related to other common diseases related to the brain, kidney or liver, so we collected them to see if any abnormality of the body may be a risk factor of dementia. As we previously mentioned, there are assessment methods

to evaluate dementia; thus, the CDR scale, the MMSE scale, the final diagnosis of what type of dementia was present, and the dementia's severity was also documented. Other personal information—such as their name, phone number, hospital history number, or address—were not included, as they are private and not related to this research.

There was clean-up and formatting work to do for the collected data, which had been stored in the database for several years. Many data cleansing methods have been widely discussed for big data analysis [22,27]. Without proper cleansing and formatting, the data will not produce insightful information. Thus, before we began the analysis, any inconsistent, duplicate, or missing data had to be fixed. Some work was performed to correct and unify the data with unreasonable values, different synonyms, or typos in order to maintain data consistency. Furthermore, it was necessary to integrate the variable categories and to merge, modify or delete the categories that were too complicated. The patient's data was removed if there were many missing values. If only a few minor values disappeared, we averaged the corresponding values of the same patients' data and filled the blank with the mean value [22,27]. There were 768 patient records used for analysis after data cleaning, composed of 292 male patients and 476 female patients.

Furthermore, it was necessary to classify the proper categories for big data analysis. In this study, the types of dementia had been classified into degenerative dementia, vascular dementia, mixed dementia, and reversible dementia. Degenerative dementia included Alzheimer's disease (AD), dementia with Lewy bodies (DLB), frontotemporal dementia (FTD), mild cognitive impairment (MCI), and Parkinson's disease dementia (PDD). Vascular dementia included depression and vascular dementia (VD) [9,28]. The other type is called reversible dementia. Because most forms of dementia are incurable, however, these are conditions that may be associated with cognitive or behavioral symptoms that can be resolved once the underlying cause is addressed [29]. There are numerous potential causes that can result in the reversible impairment of neurocognitive function in an elderly person, such as alcoholism, neurosyphilis, normal-pressure hydrocephalus (NPH), the purified protein derivative (PPD), and post-traumatic stress (PTS). The final type is "mixed dementia", which refers to a person whose brain changes when more than one cause of dementia occurs simultaneously. Alzheimer's disease and vascular dementia occurred together the most often in mixed dementia cases. The other classified variable was dementia severity. It was divided into four types: suspicious and undetermined, mild dementia, moderate dementia and severe dementia. The first type generally happened to those patients who did not have long enough observation periods after treatment, for whom the dementia symptoms appeared occasionally, or the results from the evaluating tool (CDR or MMSE) were marginal. Our experience reveals that this type of patient will very likely deteriorate to mild dementia. Gender, another variable, was divided into male and female. Education level was also a variable, and was classified from uneducated (cannot read or write), elementary school level (including not finished), and middle school and above (junior high, senior high, college, graduated school, and above). As we previously mentioned, this town is located in the countryside, and with many young people away from their families, the ways the elderly are taken care of are different. This variable is classified as stay-alone without any care, care provided by the family member (family members, spouses, or cohabitants), care from home services provided by private institutes (incudes the domestic chore services, foreign workers, foreign caretakers), and institutional care (includes daycare centers, nursing homes, long-term care institutions, maintenance institutions, and nursing institutions). Because the research showed that the living environment, specifically air quality, might be related to dementia [30], we classified the living environment as being close to a mountainous area or living away from a mountainous area. The rest of the classifications are related to the physical examination variables, and are either positive or negative. For example, positive (with) for Stroke, or negative (without) for hypertension, etc. The following tables show the statistical data collected from our self-developed customized software. Table 1 is the statistics of the number of males and females in the four severities of dementia.

**Table 1.** Gender and dementia severity statistics.

|  | Suspicious/ Undetermined | Mild | Moderate | Severe |
|---|---|---|---|---|
| Male | 121 | 59 | 59 | 53 |
| Female | 173 | 116 | 88 | 99 |
| Total | 294 | 175 | 147 | 152 |

Table 2 is the statistical result of the number of male and female patients in the four types of dementia discussed in this study.

**Table 2.** Gender and types of dementia statistics.

|  | Degenerative Dementia | Vascular Dementia | Mixed Dementia | Reversible Dementia |
|---|---|---|---|---|
| Male | 223 | 26 | 21 | 22 |
| Female | 382 | 38 | 31 | 25 |
| Total | 605 | 64 | 52 | 47 |

Table 3 shows the number of patients associated with different types of dementia-related diseases.

**Table 3.** Dementia types and the associated population statistics results.

| Dementia Type | Disease | Numbers |
|---|---|---|
| Degenerative Dementia | Alzheimer's disease | 370 |
|  | Dementia with Lewy bodies | 2 |
|  | Frontotemporal dementia | 3 |
|  | Mild Cognitive Impairment | 204 |
|  | Parkinson's disease | 26 |
| Vascular Dementia | Depression | 4 |
|  | stroke, brain hemorrhage, brain blood vessel disease | 60 |
| Mixed | Mixed Symptoms from other types of dementia | 52 |
| Reversible Type | alcoholism | 2 |
|  | Neurosyphilis | 2 |
|  | normal pressure hydrocephalus | 4 |
|  | purified protein derivative | 1 |
|  | post-traumatic stress | 38 |

The system we developed also collected the patients' educational levels, which is important information for the residents in the rural area. Many pieces of research were not focused on the patients' education levels but encouraged patients to read machines or newspapers so that they can have up-to-date topics used in their social activities. We did find that if the patients did not receive enough education, they tend to be quieter and less social in their interactions. Even though cellular phones with communication apps are popular today, many dementia patients could not utilize them because of their educational background. Table 4 shows the number of patients with three different educational distributed in the four types of dementia severity.

**Table 4.** Education level and dementia severity statistics.

| | Suspicious/ Undetermined | Mild | Moderate | Severe |
|---|---|---|---|---|
| Uneducated | 115 | 83 | 90 | 111 |
| Elementary | 103 | 67 | 36 | 28 |
| Middle school and above | 76 | 25 | 21 | 13 |
| Total | 294 | 175 | 147 | 152 |

Because the way in which the care methods of the dementia population are distributed is rarely discussed, Table 5 shows the distribution of the population with different care methods in different types of dementia.

**Table 5.** Statistical results of the care methods and types of dementia.

| | Degenerative Dementia | Vascular Dementia | Mixed Dementia | Reversible Dementia |
|---|---|---|---|---|
| None | 192 | 17 | 1 | 16 |
| By family member | 230 | 23 | 31 | 10 |
| Home Service | 89 | 8 | 8 | 13 |
| Institutional care | 94 | 16 | 12 | 8 |
| Total | 605 | 64 | 52 | 47 |

Different living areas may have different types of drinking water and different types of food, which might be the risk factors for dementia. Table 6 shows the distribution of the number of dementia patients living in mountainous areas and non-mountainous areas.

**Table 6.** Statistical results of the living area and type of dementia.

| | Degenerative Dementia | Vascular Dementia | Mixed Dementia | Reversible Dementia |
|---|---|---|---|---|
| Non-Mountainous | 212 | 27 | 20 | 18 |
| Mountainous | 393 | 37 | 32 | 29 |
| Total | 605 | 64 | 52 | 47 |

Research has shown that female patients are more prone to Alzheimer's disease than male patients [31]. Other research has shown that male patients with diabetes, stroke, hypertension, and hyperlipidemia are more likely to suffer from vascular dementia [32,33]. Furthermore, clinical experience revealed that the patient's gender and history of hypertension, hyperlipidemia, diabetes, and cerebral apoplexy are highly related to the types of dementia they develop [34,35]. These outcomes agree with the dementia data collected from the medical information system we developed. As we mentioned, this town is located in a rural area where many young adults move to nearby cities for a living. Education level and age also seem to be important factors related to dementia in the elderly.

## 4. Hypothesis and Data Analysis

The causes and factors related to the type of dementia that develops can provide medical staff with references for evaluation. They can also assist physicians in judging the type of dementia in patients, and can improve the accuracy of clinical diagnoses. Factors related to the severity of dementia can help doctors identify who are the high-risk groups and increase the probability of detecting the disease in its early stages for better treatment. Before establishing the research hypotheses, this study first excluded variables that would

interfere with the results of the analysis. CDR and MMSE scores are important bases for the evaluation of the severity of dementia, and there is a cause-and-effect relationship between these pieces of information. Therefore, the CDR score and MMSE score variables are excluded from the data. There is also a cause-and-effect relationship between care style and dementia severity. As the severity of dementia increases, it becomes increasingly difficult for patients to maintain their daily lives independently. A high-severity patient will rely on more professional care. As a result, the patients' families will choose the most appropriate method of care according to the severity of the patient's dementia. This will result in the worse-off dementia patients adopting more professional care methods. Thus, the severity of dementia and the patient's care style was also removed from the collected data.

Dementia-related diseases have been studied for a long time, and their causes, risk factors, and preventions have been widely discussed [35,36]. However, most of these research materials focus on wealthier people in the United States and European countries. Wally Ho, an experienced neurologist of this research team, hoped to analyze the present dementia situation in a small town in Taiwan based on those published research outcomes, and to compare our analysis results with the existing conclusions published for those patients abroad. To start the analysis, we followed the literature and examined the data we collected carefully. If the item did not have many cases, the item would not present any significant insightful value for analysis. We had only two cases of alcoholism, two of neurosyphilis, and one purified protein derivative; thus, the SPSS would not find anything useful by including them. The data with less than 10 cases were excluded (refer to Table 3). After that, we then set the following hypotheses:

**H1.** Different genders tend to have certain types of dementia.

**H2.** If dementia occurs, female patients tend to have severe symptoms.

**H3.** The lower the level of education, the worse the severity of dementia that might occur.

**H4.** The living residential area might be related to the severity of dementia.

**H5.** Patients' hypertension might be related to the type of dementia.

**H6.** Patients' hyperlipidemia might be related to the type of dementia.

**H7.** Patients' hyperlipidemia might be related to the severity of dementia.

**H8.** Patients' diabetes might be related to the type of dementia.

**H9.** If the patient has diabetes, then the severity of dementia tends to be more serious.

**H10.** Stroke increases the risk of vascular dementia.

**H11.** Strokes might be related to the severity of dementia.

This research adopted SPSS statistical software as a tool for big data analysis. SPSS stands for Statistical Package for the Social Sciences, and it is commonly used by researchers for complex statistical data analysis. It was originally launched in 1968 by SPSS Inc., and was later acquired by IBM in 2009. It is an easily operated tool for big data analysis. The Pearson Chi-squared test statistical method was utilized in this study [19].

*Data Analysis Results*

As we collected many variables from the self-developed system, the Pearson Chi-square test was the first tool we used to analyze the relationship between two kinds of variables [19]. This method is used to test whether the relationship between two kinds of variables is independent or related. For example, when we discuss dementia diseases, the chi-square test can be used to analyze the data of MCI (mild cognitive impairment) patients. We have found that aged males with vascular diseases and low education are at the most risk of their MCI transitioning into dementia. The test can also be used to analyze the data of patients with dementia caused by stroke, and to compare the differences between patients with and without anemia. We actually found that anemia is related to the incidence

of dementia. If the disability before a stroke can be controlled, the risk of dementia caused by anemia can be reduced. It can also be used to analyze the relationship between social activities and dementia. We found that there is a significant correlation between age and dementia. This big data analysis indicated that reading more articles, such as books or newspapers, can prevent the occurrence of dementia. The Chi-square test can also be used to analyze the relationship between depression and dementia. It was found that people with depression are more likely to suffer from dementia, and this only occurs in people without partners or living alone. With the help of the Pearson chi-square analysis, we could conclude that living with people can enhance cognitive function and delay dementia. The Pearson Chi-square test is a very useful and practical tool that is widely applied to data mining studies. Table 7 shows some of our analysis results which demonstrate the possible risk factors related to dementia.

**Table 7.** Chi-square test results for gender and dementia severity.

|  | Suspicious or Undetermined | Mild Dementia | Moderate Dementia | Severe Dementia |
|---|---|---|---|---|
| Male | 121/111.8 | 59/66.5 | 59/55.9 | 53/57.8 |
| Female | 173/182.2 | 116/108.5 | 88/91.1 | 99/94.2 |

Pearson chi-square test: $X^2$ = 3.52408, $p$ = 0.31765

Observation value/expected value.

The result shows that $p$ = 0.31765 > 0.05, indicating that the gender of the patients in this town was not related to the severity of dementia. This also indicates that gender has no critical influence on dementia severity.

The results of Table 8 shows that $p$ = 0.533142 > 0.05, indicating that the gender of the patients in this town was not related to the type of dementia seen in the patients. This chi-square analysis reveals the fact that gender has no significant effect on what type of dementia might occur in patients in this town.

**Table 8.** Chi-square test results for gender and the types of dementia.

|  | Degenerative Dementia | Vascular Dementia | Mixed Dementia | Reversible Dementia |
|---|---|---|---|---|
| Male | 223/230.0 | 26/24.3 | 21/19.8 | 22/17.9 |
| Female | 382/375.0 | 38/39.7 | 31/32.2 | 25/29.1 |

Pearson chi-square test: $X^2$ = 2.193942, $p$ = 0.533142

Observation value/expected value.

Table 9 examines the relationship between education level and dementia severity. Much of the dementia data collected in western countries did not discuss education level, as education was not an issue for other countries. However, the citizens of this small town are less educated even compared to citizens in metropolitan cities in Taiwan. Because of the general poverty in society when they were young, they lost the opportunity to be educated because they had to work for a living. Education was not very common for these elder citizens in their youth. The younger citizens with better education migrate to other cities. Whether education level is a risk factor is very rarely studied in general dementia studies. The following is the analysis result of different education levels and dementia severities from the Pearson Chi-square test.

**Table 9.** Chi-square test results of education level and dementia severity.

|  | Suspicious or Undetermined | Mild Dementia | Moderate Dementia | Severe Dementia |
|---|---|---|---|---|
| Uneducated | 115/152.7 | 83/90.9 | 90/76.4 | 111/79.0 |
| Elementary | 103/89.6 | 67/53.3 | 36/44.8 | 28/46.3 |
| Middle school and above | 76/51.7 | 25/30.8 | 21/25.8 | 13/26.7 |
| Pearson Chi-square test: $X^2 = 60.40122$ ***, $p = 3.73064 \times 10^{-11}$ | | | | |

Observation value/expected value, *** $p < 0.01$.

The results show that $p = 3.73064 \times 10^{-11}$, which is much less than 0.01. This analysis indicates that the education level of the patients in this town was related to the severity of their dementia. We discovered that the elderly patients with dementia who have a lower education level are less interested in social issues that occur daily. Dementia patients with no/low education's preferred entertainment is watching TV, which lacks interaction/conversation with others. They interact less with people because most of them could not read newspapers or books to have topics to communicate with neighbors. Quiet and loneliness are common in elderly people with dementia with limited education. This result tells us that the education level is significantly related to the severity of dementia. The less education the patient has, the more severe their dementia might grow as they grow older.

The result from Table 10 shows that $p = 2.56847 \times 10^{-42}$, which is almost zero, and is much less than 0.01. This analysis indicates that the care method for the patients was related to the severity of dementia. This result is also understandable because patients with mild dementia can take care of themselves, and most of them live with their families or do not need additional care. Patients with more severe dementia often need additional professional institutions to take care of them.

**Table 10.** Chi-square test results of care methods and dementia severity.

|  | Suspicious or Undetermined | Mild Dementia | Moderate Dementia | Severe Dementia |
|---|---|---|---|---|
| None | 134/86.5 | 48/51.5 | 21/43.3 | 23/44.7 |
| By Family Member | 133/112.5 | 80/67.0 | 50/56.3 | 31/58.2 |
| Home Service | 16/45.2 | 27/26.9 | 46/22.6 | 29/23.4 |
| Institutional care | 11/49.8 | 20/29.6 | 30/24.9 | 69/25.7 |
| Pearson Chi-square test: $X^2 = 219.5771$, $p = 2.56847 \times 10^{-42}$ | | | | |

Observation value/expected value.

Table 11 is the test results of living area and dementia severity from the analysis.

**Table 11.** Chi-square test results of living area and dementia severity.

|  | Suspicious or Undetermined | Mild Dementia | Moderate Dementia | Severe Dementia |
|---|---|---|---|---|
| Non-Mountainous | 105/106.0 | 68/63.1 | 57/53.0 | 47/54.8 |
| Mountainous | 189/188.0 | 107/111.9 | 90/94.0 | 105/97.2 |
| Pearson Chi-square test: $X^2 = 2.81991$, $p = 0.42023$ | | | | |

Observation value/expected value.

The result of Table 11 show that $p = 0.42023 > 0.05$, indicating that the living area of the patients was not related to the severity of dementia. We believed that because this town is not very big, the food, water, and air would be of similar quality. The living area is not very important for residents related to the dementia severity.

We then have the following summaries:

- The gender was not related to the occurrence of the type of dementia.
- The gender was not related to the severity of dementia.

- The education level was related to the severity of dementia.
- The care method the patients adopt was related to the severity of dementia.
- The living area was not related to the severity of dementia.

Compared to our hypotheses, H1 and H2 do not hold, but H3 does agree with our expectation. This Chi-square test method was applied to other categories, by the results of different $p$ values, and the results are summarized as follows:

- The area in which the patient lived was not related to the severity of dementia.
- The patients with or without hypertension were not related to the type of dementia.
- The patients with or without hyperlipidemia had a significant correlation to the type of dementia. The Chi-square test shows $p = 0.002$, which indicated a strong relation between hyperlipidemia and dementia.
- The patients having or not having hyperlipidemia as not related to the severity of dementia. The Chi-square showed that the $p$-value is 0.16 > 0.05.
- Whether a patient had diabetes or not was not related to the type of dementia, as its $p$-value was 0.46 > 0.05.
- There may be some correlation between diabetes and the severity of dementia. The Chi-square test showed that $p = 0.068$; it was greater than 0.05, but less than 0.1. This might indicate that the population who has diabetes might be related to the severity of dementia that develops. We believe that this may be because patients with diabetes are affected by dysglycemia, resulting in brain nerve damage, structural changes, or vascular lesions, leading to an increased risk of dementia. In addition, patients with diabetes often also have obesity, and middle-aged obese people are prone to dementia.
- There may be some correlation between whether a stroke occurred in the past and the type of dementia that develops. The Chi-square test showed that the $p$-value is 0.097; it is greater than 0.05, yet less than 0.1. Hemorrhagic stroke is often a cerebrovascular disease, which has a strong relation to vascular dementia.
- Whether patients had had a stroke or not was not related to the severity of dementia. Its $p$-value was 0.103, which is greater than 0.1.

From the above summaries, we know that H4, H5, H7, H8, and H11 don't hold, but H6, H9, and H10 do agree with the hypothesis, even though the statistical evidence for H9 and H10 was not particularly strong. This might be because the samples we collected for patients with diabetes and who suffered strokes were not significant enough. We concluded that whether the patient had had a stroke or not and whether they had had hyperlipidemia or not both influenced the type of dementia the patient developed. Diabetes mellitus is a risk factor for the severity of dementia.

## 5. Discussion

Dementia is a global malady with growing ramifications for society, the economy, and families. Many individuals have benefited from this cooperative research. The contributions of this study can be summarized as follows. The hospital was built to serve the economically disadvantaged citizens of this community. A single visit to this hospital costs about $5 to $8, in addition to medications and possible examinations. Some may be able to waive the cost if it is combined with National Health Insurance. The dementia patients we gathered are concentrated among the population of this small town and the surrounding regions due to the hospital's geographical position. Although urban residential areas may hold more dementia patients, they are not included in the scope of this study. Furthermore, studies on dementia risk factors have been conducted with relatively few studies focusing on a specific group of patients. As a result, the trustworthiness of this study for the local residents of this area is quite high. We discussed our findings with academic institutions and relevant government agencies, and local governments started to build more long-term care facilities and develop daycare programs based on our findings. For instance, our research revealed a connection between education level and dementia severity. Because of poverty, loneliness, and a lack of social engagement, some dementia sufferers' conditions have progressively gotten worse. The development of the daycare facility currently allows people

with dementia to visit during the daytime and engage in carefully developed interactive activities, such as drawing classes, folk dance programs, or chess play. Furthermore, our research showed that dementia progresses more quickly in people residing in care facilities. The faculty of the care facility can notify the patient's family ahead of time if the patient's symptoms worsen. The family may anticipate and plan for all of the necessary safeguards ahead of time. This is extremely beneficial to the patients and their families. Through mutual discussion, software development engineers learned how to collaborate with hospitals to produce effective integration software.

This research inevitably has limits and downsides. As previously stated, the goal of this study is to identify potential dementia risk factors among people residing near this hospital. For the people of this community, our results are quite accurate. However, such findings may not be applicable to the older populations in other metropolitan regions. In reality, cities have huge populations; therefore, there should be more people suffering from dementia. If more samples can be included, the accuracy of the results will be more comprehensive and determined. However, due to geographical limitations as well as patient privacy and regulatory requirements, this study is unable to acquire patient data from other metropolitan hospitals for analysis. The other disadvantage is associated with the software system. Even though we created a specialized medical information system that can effectively store and collect information, different hospitals use different databases, and our system cannot be converted simply. In addition, our approach can only be utilized on Microsoft Windows; it will not function on other operating systems (IOS, Android, or Linux). Another limitation of this study is that only qualitative analysis of big data is employed; there is little quantitative analysis. The Chi-square test may establish whether there is a relationship between two variables, but what two categories of variables are connected requires more trials.

## 6. Conclusions

Several governments are focusing on dementia issues to provide reasonably priced medical care and a high standard of living for their aged population. Healthcare is currently one of the most important businesses around the world. Much time and energy have been spent on dementia research. The causes of dementia are well known, and treatments have been updated. However, the conclusions of dementia research may not fully apply to all people worldwide because living habits, people's affluence, living quality, food preferences, air pollution, drinking water quality, and education level may vary greatly from country to country. It is important to acquire local dementia data in order to analyze and compare that to published information so that the best care policies can be made for local individuals with dementia. Statistical approaches and big data investigations are commonly used to analyze the collected data, but accurate data acquisition is the most difficult prerequisite. Most hospitals have limited financial budgets, and as a result, the physicians in different departments share the same diagnosis software system. The physician writes down the diagnosis and the patient's chief complaints in text, and the statements are then stored in the database. Because physicians have their own writing styles and abbreviations, these statements could vary dramatically, and mining of the keywords of dementia from all of the statements in a hospital is inefficient. Furthermore, extra effort is required to manipulate the data to fit the analysis software format. Some medical facilities have sensed this difficulty and outsourced this job to software companies. However, without the involvement of a knowledgeable neurologist, the outsourced software developed from an engineer's point of view is difficult to use and may lack some basic neurologic functionalities. Integrating numerous medical systems to provide effective diagnosis and treatment and appropriate medicines for dementia patients therefore becomes a critical issue, according to research [37]. When new technologies are invented, hospitals are presented with different and new medical systems; however, efficiency and productivity issues arise if the systems are not completely connected. Nursing staff, as well as other hospital personnel, may have to work even harder to adjust to the unintegrated systems. Not only does this increase nurse

burnout, it also jeopardizes patient safety. The customized system used in this study was created with the help of a physician who was familiar with clinical procedures and requests, and it was modified based on hospital personnel feedback to improve system integration.

This self-developed program had far more medically professional and specifically tailored features than the widely used general diagnosis system currently in use in many hospitals. For patients with dementia, the government currently issues a certificate of disability. With the certificate, the patient may make insurance claims or may obtain financial aid or social welfare from the government. Most medical facilities utilize the CDR or MMSE to evaluate patients, but the questionnaires for the CDR and MMSE can be obtained from the internet. Some people take advantage of this and answer CDR/MMSE questions deceptively in order to obtain claims or disability certificates. Our customized system can access the Picture Archiving and Communication System (PACS) database to load the EEG image (or MRI/CT images, if available) to see whether the patient actually suffers from dementia; even the scale of CDR indicates the disease. This unique function helps to prevent any illegal deception. It was also found through the SPSS analysis that educational level was crucial for the local residents in this town. We found that uneducated patients over the age of 69.51 are at high risk of dementia, and this has been a very helpful unique contribution to this study. We also found that hyperlipidemia was related to the type of dementia patients developed. Patients with hyperlipidemia had a higher percentage of reversible dementia. We also found that education level and diabetes status were related to the severity of dementia, which agrees with both the literature and our hypothesis. Patients with lower education levels and diabetes mellitus tended to suffer a higher severity of dementia. Logistical regression was then used to further explore the age dividing points of patients with different educational levels and those with or without diabetes. This is important information because it can be used to warn high-risk groups about potentially developing dementia when the time approaches. Among all of these variables, we found that patient age and education level have the greatest influence on whether patients would develop dementia. Age is not a surprising factor, as we tend to develop dementia when brain function becomes vestigial as we grow old, but education level is not a widely discussed factor. We are confident that this physician-focused medical informatics system will make dementia exams a daily standard practice and enhance healthcare sustainability. It will be also an effective integration appropriate for senior healthcare. This work has collected more than seven hundred and sixty records on patients with dementia for analysis, and has proven itself as an efficient system to improve treatment for patients with dementia-related illnesses.

Using this tailored medical information system to continuously collect information from dementia patients will be a key responsibility in the future. A large number of samples can increase the data-mining results' accuracy. Other statistical approaches will be used to examine the data, and we believe that quantitative analysis might produce insightful information that will help patients' health. Seeking collaboration across hospitals to exchange research results and provide mutual assistance will also be a future research focus.

**Author Contributions:** Conceptualization, H.-H.H., J.-J.L. and T.-Y.Y.; methodology, H.-H.H., J.-J.L. and T.-Y.Y.; software, H.-H.H. and T.-Y.Y.; validation, J.-J.L., J.-Q.G. and T.-Y.Y.; formal analysis, J.-J.L., J.-Q.G. and T.-Y.Y.; investigation, J.-J.L. and J.-Q.G.; resources, H.-H.H. and T.-Y.Y.; data curation, J.-J.L., J.-Q.G. and T.-Y.Y.; writing—original draft preparation, J.-J.L. and T.-Y.Y.; writing—review and editing, H.-H.H. and T.-Y.Y.; visualization, H.-H.H. and T.-Y.Y.; supervision, H.-H.H. and T.-Y.Y.; project administration, H.-H.H. and T.-Y.Y.; funding acquisition, H.-H.H. and T.-Y.Y. All authors have read and agreed to the published version of the manuscript.

**Funding:** This research received no external funding.

**Institutional Review Board Statement:** Not applicable.

**Informed Consent Statement:** Not applicable.

**Data Availability Statement:** Data available on request due to restrictions on patients' privacy.

**Conflicts of Interest:** The authors declare no conflict of interest.

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
