# Peer review of "An Empirical Study for Senior Citizens Using a Customized Medical Informatics System for Dementia Diagnosis and Analysis"

_sustainability, doi:10.3390/su14159064_

Round 1
Reviewer 1 Report
1. For introduction section , I highly suggested that author should re-write this section, including background, present reviews for data mining, major methods and drawback of presented data mining method. Some sentences, for example line68-70, is Encyclopedia-like knowledge, which should be not mentioned in academic literature.
2. For methods and implementation section, could author provide a flow-chart-like diagram to illustrate this? It likes that authors use too many words.
3. author did not provide enough mathematics-related backgrund. This software is how to analysis these data? Any explanation?
Reviewer 2 Report
In this manuscript, the authors developed a customized medical information system to collect and track dementia patients. The following opinions were formed by examining the article in detail:
1. More recent related studies (published in 2020-2021) can be added as references.
2. The manuscript should include a paragraph that describes the manuscript’s structure as the last paragraph of the introduction, e.g., “This paper is structured as follows...”
3. The novelty and main contributions of this study should be stated as one paragraph in the “Introduction” section.
4. Possible future work(s) should be given to guide the readers.
5. The manuscript should be reorganized. Although there is no heading with the number 2, there is a sub-heading with the number 2.1.
6. Some abbreviations are used in the text without giving their expansion. For example: CDC, DBMS, VBA etc. The authors should write that "these abbreviations stand for what".
7. An example (structure of the proposed method) should be given to make the proposed model easy to understand.
Reviewer 3 Report
The aim of the manuscript is to help in understanding how dementia developed in rural areas in Taiwan. Customized medical information system was developed and presented. Data records collected on 768 patients were used for data analysis. The authors used big data technology and data mining approaches. The authors proposed 11 hypotheses, while the test was performed by using Pearson chi-square test. Results are useful for decision-makers for defining policies for elder citizens.
The manuscript is well structured, containing introduction to the topic, description of developed in-house information system, description of the process for collecting and analyzing data, and finally discussion of the results based on the proposed research hypotheses. The author uses relevant and up-to-date references.
However, there is a need to improve the manuscript in the following ways:
1. The research findings and the benefits of the presented research should be clearly stated in the abstract.
2. The implications and benefits for the community and the researchers should be discussed, in the conclusion section or in a separate section before conclusions.
3. Potential further work directions should be discussed at the end of the Conclusions section.
Reviewer 4 Report
This manuscript investigated dementia in the aged population, and big data analysis was applied to determine risk factors and the relationship with disease severity.
I have a few suggestions and questions below:
1. Abstract: In the final part of the abstract, please briefly summarize the important findings of this study, instead of just describing useful findings obtained.
2. Line 118-136: The advantages of big data approaches seem to belong to the introduction part rather than the methods part.
3. Discussion: Considering not all readers of Sustainability in medical care fields understand well with the big data approach, could you please add a short paragraph to provide possible drawbacks and limitations of the methods with this study?
Reviewer 5 Report
Please find attached my reviews for this paper.

Round 2
Reviewer 1 Report
Author have well answered my questions, but there are several tiny mistake. Please carefully check it as following questions.
A. line26, please correct these words “Thanks to advances in technology” to “benefit from advances in technologies”. This is academic paper, not “Religion Forum”.
B. Line30-31 is similar with line 36-37, please simplify this.
D. Line83-84, references missing
E. Please cite these references in line103-104 to support your points.
1. Le T T, Andreadakis Z, Kumar A, et al. The COVID-19 vaccine development landscape[J]. Nat Rev Drug Discov, 2020, 19(5): 305-306.
2. Cao Y. The impact of the hypoxia‐VEGF‐vascular permeability on COVID‐19‐infected patients. Exploration. 2021, 1(2): 20210051.
3. Mahmud M A P, Tat T, Xiao X, et al. Advances in 4D‐printed physiological monitoring sensors. Exploration. 2021, 1(3): 20210033.
4. Yasuda S, Miyamoto Y, Ogawa H. Current status of cardiovascular medicine in the aging society of Japan[J]. Circulation, 2018, 138(10): 965-967.
Reviewer 2 Report
The revised paper has addressed my comments and I do not have additional concerns.
Reviewer 5 Report
Many thanks for the authors' effort.
I have no further comments on the revised paper.
Author Response
Please see the attachment.

This manuscript is a resubmission of an earlier submission. The following is a list of the peer review reports and author responses from that submission.